# A Graphene-Based Bioactive Product with a Non-Immunological Impact on Mononuclear Cell Populations from Healthy Volunteers

**DOI:** 10.3390/nano14231945

**Published:** 2024-12-04

**Authors:** María del Prado Lavín-López, Mónica Torres-Torresano, Eva María García-Cuesta, Blanca Soler-Palacios, Mercedes Griera, Martín Martínez-Rovira, José Antonio Martínez-Rovira, Diego Rodríguez-Puyol, Sergio de Frutos

**Affiliations:** 1Graphenano S.L., 30510 Yecla, Spain; pradolavin@graphenano.com (M.d.P.L.-L.); mercedesgriera@graphenano.com (M.G.); martin@graphenano.com (M.M.-R.); jose@graphenano.com (J.A.M.-R.); 2Department of Immunology and Oncology, National Center for Biotechnology, Consejo Superior de Investigaciones Científicas (CSIC), 28049 Madrid, Spain; monica.torres@iisgm.com (M.T.-T.); emgarcia@cnb.csic.es (E.M.G.-C.); bsoler@cnb.csic.es (B.S.-P.); 3Department of Medicine, Universidad de Alcalá, Nephrology Service at Hospital Príncipe de Asturias, Instituto Ramon y Cajal de Investigación Sanitaria, Fundación Renal Iñigo Álvarez de Toledo, 28871 Alcalá de Henares, Spain; diego.rodriguez@uah.es; 4Department of Systems Biology, Universidad de Alcalá, Instituto Ramon y Cajal de Investigación Sanitaria, Fundación Renal Iñigo Álvarez de Toledo, 28871 Alcalá de Henares, Spain

**Keywords:** graphene, carbon nanofiber, leucocytes, lymphocytes, natural killers, monocytes, macrophages, toxicity, phagocytosis, activin A

## Abstract

We previously described GMC, a graphene-based nanomaterial obtained from carbon nanofibers (CNFs), to be biologically compatible and functional for therapeutic purposes. GMC can reduce triglycerides’ content in vitro and in vivo and has other potential bio-functional effects on systemic cells and the potential utility to be used in living systems. Here, immunoreactivity was evaluated by adding GMC in suspension at the biologically functional concentrations, ranging from 10 to 60 µg/mL, for one or several days, to cultured lymphocytes (T, B, NK), either in basal or under stimulating conditions, and monocytes that were derived under culture conditions to pro-inflammatory (GM-MØ) or anti-inflammatory (M-MØ) macrophages. All stirpes were obtained from human peripheral mononuclear cells (PBMCs) from anonymized healthy donors. The viability (necrosis, apoptosis) and immunological activity of each progeny was analyzed using either flow cytometry and/or other analytical determinations. A concentration of 10 to 60 µg/mL GMC did not affect lymphocytes’ viability, either in basal or active conditions, during one or more days of treatment. The viability and expression of the inflammatory interleukin IL-1β in the monocyte cell line THP-1 were not affected. Treatments with 10 or 20 µg/mL GMC on GM-MØ or M-MØ during or after their differentiation process promoted phagocytosis, but their viability and the release of the inflammatory marker activin A by GM-MØ were not affected. A concentration of 60 µg/mL GMC slightly increased macrophages’ death and activity in some culture conditions. The present work demonstrates that GMC is safe or has minimal immunological activity when used in suspension at low concentrations for pre-clinical or clinical settings. Its biocompatibility will depend on the dose, formulation or way of administration and opens up the possibility to consider GMC or other CNF-based biomaterials for innovative therapeutic strategies.

## 1. Introduction

Graphene-based carbon materials (GBMs) are receiving considerable attention for multiple emerging biomedical applications, including tissue engineering. Their large surface area and active physicochemical properties promote bio-functional activities in cellular routes that affect adhesion, spreading, proliferation and even cell differentiation [1]. GBMs share the common feature of containing graphene, but they differ in terms of their structural characteristics, including their size, shape, thickness and surface modifications. These materials can range in size from a few to several hundred nanometers and are thus categorized based on their structural diversity. Some GBMs have been extensively studied to be potentially useful in biological applications, such as fiber-like shaped carbon nanofibers (CNFs) and carbon nanotubes (CNTs) or graphene oxide (GO). Between those, CNF-based materials have some advantages over CNTs or GO, such as larger scale manufacturing at a lower price and validated biomedical compatibility and uses [2,3,4,5,6,7,8,9,10]. However, studies about CNFs’ immunological activity are limited compared to other GBMs [11,12,13,14,15,16,17,18,19]. 

Each specific GBM must be carefully evaluated to be biocompatible, defined as the absence of harmful effects on living organisms, before its application in biological systems. Several works have been performed either in vitro or in vivo to assess the safety of different GBM formulations; however, the results are not conclusive since multiple factors are implicated: (a) the ulterior characteristics of each particular GBM, such as the size, shape, charge, agglomeration state, number of graphene layers, surface functionalization, manufacturing and purification procedures, and (b) the experimental design, which depends on the cell type or model used, the administration pathway, the time of GBM exposure and the dosage [20,21,22,23,24]. Most studies related to GBM toxicity analysis evaluate relevant pathological responses triggered by the compounds. However, the current state of the art demands that exogenous materials must be as innocuous as possible when entering into contact with a living organism, thus biocompatible, to ascertain the safety of the material in clinical settings. The concept of biocompatibility goes beyond toxicity, as subtle changes in immune system activity may determine biological responses that induce maintained pathological changes in the organism in contact with the material.

In a previous study, we investigated the functional application and compatibility of a CNF-based material manufactured and patented by Graphenano Medical Care S.L. (WO 2020/016319) with no registered trademark and named here as GMC using the firm name [25]. We demonstrated that GMC can be safely used in concentrations lower than 150 µg/mL in in vitro and in vivo models when administrated as a suspension, while having biological effects on the functionality and trans-differentiation of systemic cells. In a pre-clinical approach, the topical administration of GMC in an overweighted/obese rodent model was apparently biocompatible since no changes in systemic toxicity parameters were observed. The lack of toxicity was confirmed using cultured adipoblast-like pluripotent cells, as well as other systemic cellular models. The bio-functional effect of GMC in the underlying subcutaneous white adipose tissue, and confirmed in cultured adipocytes, was the ability to reduce triglycerides’ content through the interaction with the extracellular matrix receptor integrin beta 1 and the downstream mediator integrin-linked kinase (ILK). We demonstrated that GMC administered in those conditions could be used as an innovative anti-obesogenic strategy. The present study expands upon this work by examining the effect of GMC on immunoreactivity and cell viability in more detail, focusing on lymphocytes and macrophage progenies obtained from the peripheral blood mononuclear cells (PBMCs) of anonymous healthy volunteers, using a standardized flow cytometry gating strategy and data analysis [14,26,27]. Pro-inflammatory cytokine expression was also investigated in the cultured human monocyte cell line THP-1. This is the first complete study to elucidate the effects of a CNF-based biomaterial on PBMC subpopulations. 

## 2. Methods

### 2.1. GMC Preparation and Characterization

High-surface area CNFs supplied by Graphenano S.L were processed using methods previously described in the literature, subjected to purification and biocompatibilization processes [17,25] by treating them with a strong acid solution to remove impurities. After washing them with distilled water until they reached a neutral pH, CNFs were subjected to a standard depyrogenization process at 400 °C under an inert atmosphere to remove endotoxins. The product particle size was reduced and controlled through an exfoliation process and filtration system. Raman spectroscopy was performed using a Senterra spectrometer with a grating of 600 lines per mm and a laser wavelength of 532 nm at a power level of less than 1 mW laser power level (Bruker Española, Madrid, Spain) [25]. Crystallographic X-ray (XPS) diffraction parameters were determined using a Phillips PW-1711 diffractometer with CuKa radiation and a 1.5404 Å wavelength. The interlaminar distance was calculated with Braggs’s equation λ/2·sen (θ), where λ is 1.5404 Å long-wave radiation and θ is the diffraction angle, the stacking aromatic structure and the orientation of graphene layer planes, respectively [25]. The crystalline size in the perpendicular plane or the crystal stack height of graphene layers was obtained with Scherrer’s equation Lc = Kc·λ/FWHM·cos(θ), where Kc is the carbonous constant (=0.9) and FWHM is the Full Width at Half Maximum of the corresponding diffraction peak, in rad units [25]. Scanning Electron Microscopy (SEM) images were obtained using Zeiss GeminiSE500 equipment with a thermal field emission, an acceleration voltage from 0.02–30 KV and a magnification from 50 to 2,000,000. High-Resolution Transmission Electron Microscopy was performed using FEI Tecnai G2 F20 S-TWIN equipment [25].

### 2.2. Leucocyte Progenies’ Purification, Culture Conditions and Analysis

THP-1 human monocyte cells (TIB-202, ATCC, distributed by LGC Standards, Barcelona, Spain) were grown with RPMI-1640 plus 10% heat-inactivated fetal bovine serum (Fisher Scientific, Madrid, Spain) and deprived prior to GMC treatment. 

Buffy coats from a total of 6 anonymous healthy donors were kindly donated by the blood bank at the Madrid Transfusion Centre of the Autonomous Madrid Government to the National Centre for Biotechnology (CSIC, IRB approval number 280508). The procedures to obtain blood and buffy coats followed the standards of transfusion institutions, including obtaining informed consent from the donors. Human peripheral blood mononuclear cells (PBMCs) were isolated from donated buffy coat samples using Lymphoprep separation (Nycomed Pharma AS, Oslo, Norway). Leucocyte progenies were obtained from PBMC pools cultured in the following conditions and either stimulated (activated) with the specific stimuli or maintained in basal conditions at the indicated times [14,27]. All the PBMC progenies were cultured at a density of 10^6^ cells/mL maintained in an RPMI-1640 medium supplemented with 10% FBS (all from Fisher Scientific, Pittsburgh, PA, USA), with or without the specific stimulants, and placed in a 5% CO_2_ atmosphere maintained at 37 °C. After the treatments, cultured progenies were incubated with conjugated antibodies against the following surface markers: CD3-FITC, CD23-FITC, CD56-PE, CD56, CD56^lo^, CD86 and CD107a/LAMP1-APC (Biolegend, San Diego, CA, USA); CD25-PE and CD69-PE (Becton Dickinson, Franklin Lakes, NJ, USA); and CD19-PC5 and CD14-PE (Beckman Coulter International, Nyon, Switzerland). Samples were analyzed using a Gallios Flow Cytometer (Cytomics FC 500) and FlowJo software v10 (Beckman Coulter International, Nyon, Switzerland). PBMCs were stimulated with 20 ng/mL IL2 and 0.5 μg/m PHA-M medium for 24 h and switched to medium with only IL2 for another 72 h to obtain T blast cells (active) or maintained unstimulated to study T-lymphocytes in basal conditions. B-lymphocytes were purified from PBMC pools through magnetic cell sorting using the commercial kit EasyStep Human CD19 Positive Selection Kit II (Stem Cell Technologies, Vancouver, BC, Canada) and cultured afterwards under stimulating (active) conditions with 1μM ODN 2395-CpG, 1 mg/mL IgG + IgM and 100 U/mL IL4, or maintained unstimulated under basal conditions (Merck, Darmstadt, Germany). To study NK progenies, PBMCs were cultured for 7 days with or without GMC. Thereafter, cells were co-cultured with a target cell line for NK cells, K562 (CCL-243, ATCC, distributed by LGC Standards, Barcelona, Spain), for 2 h at a 5:1 ratio to activate NK progenies (around 10% of the total PBMCs) or maintained under the same conditions but without a target. After the treatments, cells were stained with specific antibodies for NK progenies and the NK degranulation (activity) marker LAMP1. Monocytes were purified using a commercial kit based on magnetic cell sorting, CD14 MicroBeads (Miltenyi Biotec, Bergisch Gladbach, Germany), and subjected afterwards for 7 days to specific differentiating factors added to the culture medium every 2 days. For pro-inflammatory macrophage (GM-MØ) differentiation, granulocyte macrophage colony-stimulating factors (GM-CSFs, 1000 U/mL) was added, while for reparative, anti-inflammatory macrophage phenotype differentiation (M-MØ), a macrophage colony-stimulating factor (M-CSF, 10 ng/mL) was added (ImmunoTools, Friesoythe, Germany). Treatments with GMC or the control (untreated) were performed during or after the purification steps at the indicated times. After the treatments, cells were dyed with specific antibodies and reagents and analyzed using flow cytometry. The presence of the macrophage pro-inflammatory marker activin A released from GM-MØ macrophages was determined in the cultured cells’ supernatants after the differentiation/treatment procedures (7 days) by using the commercial ELISA kit Quantikine immunoassay (R&D Systems, Minneapolis, MN, USA) [14,26,27].

### 2.3. Cell Cycle Profiles 

After each treatment, cells were washed with cold phosphate-buffered saline (PBS) and resuspended in 50 µL of detergent (DNA-Prep Reagent Kit; Beckman Coulter, Brea, CA, USA) containing 10 ng/mL propidium iodide (PI, DNA-Prep Reagent Kit; 30 min, 37 °C). Cell cycle phases were analyzed using flow cytometry with a Beckman Coulter FC500 flow cytometer and the results were expressed as the percentage of stained cells [14,26,27]. 

### 2.4. Apoptosis and Necrosis Analysis 

After each treatment, cells were washed with cold PBS and stained with propidium iodide (PI), which intercalates to DNA in cells with compromised plasma, and fluorescein isothiocyanate (FITC)-Annexin V (Merck, Darmstadt, Germany), which binds to the phosphatidyl serine that translocates to the outer leaflet of the cells during early apoptosis (Annexin V+). Dyed cells were analyzed using flow cytometry to differentiate between viable, apoptotic (early or late) and necrotic cells. The percentage of apoptotic (Annexin V+), necrotic (Annexin V−/PI+) and viable cells (Annexin V−/PI−) was determined based on the flow cytometry analysis [14,26,27].

### 2.5. Reverse Transcription–Quantitative Polymerase Chain Reaction (RT-qPCR)

After the experiment, total RNA was extracted from cells or tissues collected from fasting mice. Equal amounts of RNA were transcribed to cDNA with a HighCapacity cDNA RT Kit and 10 ng of cDNA was amplified using kits for qPCR. TaqMan gene expression assays were used to quantify Interleukin-1β (IL-1β, Mm00434228_m1) and β-actin (Mm01205647_g1). All products and equipment used were from Thermo Fisher Scientific (Madrid, Spain). Amplification values were normalized to endogenous β-actin and the relative quantification was determined using the 2^−ΔΔCT^ method [25,28,29]. 

### 2.6. Statistical Analysis

For comparisons between two groups, Student’s *t*-test was applied, while Friedman analysis was used for comparisons involving more than two groups, paired and non-parametric, followed by Dunn’s post hoc test for multiple comparisons. The study had a statistical power of 80–85%, with a 95% confidence level. Statistical analyses were performed using GraphPad Prism 8 Software. Differences were considered statistically significant at a *p*-value of less than 0.05. 

## 3. Results

### 3.1. GMC Characterization and PBMC Viability After GMC Treatment

During the production and purification of GMC, raw material was subjected to a strong acid solution and high temperatures in an inert atmosphere to avoid contaminants. Contaminants such as metal impurities or other substances, such as pyrogens derived from bacteria, fungi and viruses, may be present during the manufacturing of GBM and potentially evoke toxicity or immunological activity in cultured leucocytes [11,15].

Inductively Coupled Plasma Mass Spectrometry (ICP-MS) was used to determine metal traces. [Si] and [Ni] concentrations decreased to 5608 ± 38 and 6598 ± 153 µg per g of dried GMC (mean ± SEM), respectively. The rest of the metals (Cu, Zn, Fe, Ti, etc.) were present at less than 300 µg per g of GMC. Taking into consideration that GMC would be used for experimental approaches in a suspension at very low concentrations (20 to 60 µg GMC per mL), the remaining metal impurities in the suspension would be diluted at concentrations of ng/mL and would therefore be considered to be harmless. Figure 1A shows a representative High-Resolution Transmission Electron Microscopy (HR-TEM) image of agglomerated non-suspended dry GMC, while Figure 1B,C show representative Scanning Electron Microscopy (SEM) images of single GMC fiber-like structures after their dispersion into the cultured media used in the experimental settings. It can be observed that the fibers of GMC have an average length of up to hundreds of nm and diameter of tens of nm, which corroborates our previously observed size of GMC using a Dynamic Light Scattering technique [25]. Figure 1D shows the Raman spectra present in GMC, with two main peaks observed, one peak at 1350 cm^−1^ which corresponds to the D-band, which arises from structural defects in the sp^2^ carbon lattice, and another peak at 1580 cm^−1^, which represents the G-band, associated with the in-plane stretching vibrations of sp^2^ carbon atoms in graphitic structures. The intensity peak ratio for these two peaks (ID/IG) is commonly employed as a quantitative metric to assess the degree of structural disorder or defect density. The ratio of GMC was approximately 0.9, which suggests that the material has a moderate level of graphitization with a balance between ordered (graphitic) and disordered (amorphous or defective) carbon phases [7,10,25,30,31]. In addition, the deconvolution of the Raman spectra reveals the appearance of two additional peaks, the D3 (or D″) peak near 1500 cm^−1^, caused by heteroatoms and sp^2^ carbon atoms located in defects and in the amorphous phase, and the D4 (or I) peak near 1180 cm^−1^, which corresponds to C–H covalent bonds [30,31].

Figure 1E shows the high-resolution spectrum of the carbon region (C1s) obtained using X-ray photoelectron spectroscopy (XPS). The atomic % of carbon and oxygen is 79.36 and 2.51, respectively. The figure shows that GMC is composed mostly of sp2-type carbon bonds (69.63% of the bonds) with a small contribution from other C-to-O bonds (% of C–O of 9.22; C=O of 5.68; O–C=O of 3.15). These values have been obtained by using Gaussian–Lorentzian peak fitting combined with a normalized peak area model to identify and quantify the carbon and oxygen groups [31].

Our previous work studied the bio-functional effects of GMC used as a suspension on cultured cells when used in the order of tens of µg/mL, as explained in the Introduction Section [25]. Therefore, we studied the viability of primary cultured PBMCs, isolated from the buffy coats of healthy donors and treated with a range of GMC concentrations from 0 (control) to 20 or 60 µg/mL for 24 h at 37 °C. Table 1 shows the % of PBMCs during different cellular phases (G0/G1, G2/M, S), apoptosis and necrosis based on the flow cytometry analysis. GMC was harmless in terms of viability in cultured PBMCs, since no significant differences between the experimental groups were observed. Similar results were obtained when PBMCs were treated with GMC for longer periods, 48 and 72 h.

### 3.2. Immunological Activity of Lymphocytes and NK Progenies During GMC Treatment

To study the immunological activity of lymphocytic progenies left untreated and treated with 20 and 60 µg/mL GMC, isolated PBMCs from healthy human donors were cultured to obtain different lineages and subpopulations, either in a steady state or under activation conditions, by using either specific culture conditions or isolation kits, as detailed in the Methods Section and herein.

Once PBMCs were treated, the different subpopulations were stained using specific antibodies and analyzed using flow cytometry. Figure 2A shows the count (in %) of live T-lymphocytes (CD3+) that expressed markers of activation (CD69 and CD25) after the treatment either with PHA-M + IL-2 (stimulated conditions) or without (unstimulated conditions). Stimulation increased the count of active T-lymphocytes compared to those that were unstimulated, as expected. No statistically significant differences were observed between controls and the different GMC treatments after 24 h, neither in the unstimulated subpopulations, nor in the stimulated ones. 

Figure 2B shows the % of purified B-lymphocytes (CD19+ cells) that expressed markers of activation (CD23, CD69 and CD86) that were unstimulated or treated with ODN 2395-CpG + IgG +IgM + IL-4 (stimulated conditions). Stimulation increased the count of active B-lymphocytes compared to those that were unstimulated, as expected. No statistically significant differences were observed between controls and the different GMC treatments after 24 h, neither in the unstimulated subpopulations, nor in the stimulated ones. Similarly, stimulated or unstimulated T- or B-lymphocytes treated with GMC for longer periods, either 48 h or 72 h, did not show differences between groups. Altogether, these results suggest that in the case of both T and B cells, GMC is unable to activate unstimulated cells or modulate the activation status of those that have been previously activated.

To study the NK cell family, PBMCs were cultured for 7 days in the absence or presence of GMC without the addition of any exogenous stimulus. Afterwards, the capacity of NK cells to degranulate was analyzed by performing a co-culture for 2 h with K562 cells, a target for NK cell degranulation. As a control, cells with no target were incubated in the same conditions. Figure 2C shows the % of NK cells (CD3−, CD56+) that were degranulating (LAMP+). As expected, the percentage of degranulating NK cells increased when K562 cells were present, compared to cells that were not incubated with a target cell line. No statistically significant differences were observed between controls and the different GMC treatments within each NK subpopulation, suggesting that GMC is not able to modulate the capacity of NK cells to respond to target cells. The count of the NK-T subpopulation (CD3+, CD56+) was very low and differences between GMC treatment groups were also insignificant.

### 3.3. Viability and Immunological Activity of Cultured Human Monocytes THP-1 Cell Line

Figure 3 shows that 20 µg/mL GMC for 24 h, the bio-functional concentration for cultured adipocytes without deleterious effects that we previously defined [25], did not affect the viability of the human monocyte cell line THP-1, nor increased the mRNA expression of an inflammatory cytokine, IL-1β, which is used as a marker of THP-1 inflammatory activity [28]. TGF-β1 treatment was used as a positive control, which increased IL-1β expression, as expected [29].

### 3.4. Viability, Immunological Activity and Phagocytosis During GMC Treatment of GM-MØ Macrophages 

Monocytes (CD14+) were isolated from human PBMCs and cultured under specific culture conditions to obtain either pro-inflammatory (GM-MØ) or anti-inflammatory (M-MØ) macrophages for a total of 7 days (1000 U/mL GM-CSF or 10 ng/mL M-CSF in the culture media, respectively) [14,26,27]. Cells were co-treated with GMC (10, 20 or 60 µg/mL) or kept untreated (control) during or after their macrophage differentiation process, as detailed in the Methods Section and herein.

Under these conditions, we studied necrosis, apoptosis and the release of activin A. We used Annexin V and PI staining to determine necrosis (Annexin V− and PI+ staining) and apoptosis (Annexin V+, PI−). To measure pro-inflammatory markers, activin A levels were measured in the supernatant from the cultured cells. Like the above-mentioned pro-inflammatory cytokines TGF-β or IL-1β, activins are considered a crucial modulator of the immunological response, especially during the macrophage’s polarization to the pro-inflammatory phenotype [26,27]. 

Figure 4A,B show necrosis and apoptosis in fully differentiated GM-MØ after 24 h of GMC treatments. When compared to the control, GMC promoted a weak increased tendency in a dose-dependent manner for both necrosis and apoptosis, being statistically significant when using the highest dose, 60 µg/mL. Figure 4C shows activin A levels in the cultured supernatant from fully differentiated GM-MØ exposed to GMC for 24 h. Compared to control values, GMC treatment did not affect activin A production. 

Figure 4D,E shows the necrosis, apoptosis and activity of GM-MØ treated with GMC for 7 days in the culture, during their macrophage differentiation period. GMC did not affect necrosis or apoptosis at any dose compared to the control. On the other hand, Figure 4F shows that activin A levels were slightly increased in a GMC dose-dependent manner, being statistically significant when using the highest dose, 60 µg/mL.

Functionally, macrophages phagocyte pathogens and other agents [14,26,27]. We therefore studied the side/forward scatter analysis of flow cytometry and performed contrast phase microscope photography to study the morphological and size changes that could happen during GMC phagocytosis by the different subpopulations of macrophages. Figure 5A shows a representative graph of a flow cytometry analysis from differentiated GM-MØ left untreated or treated with 20 µg/mL GMC for 24 h. The graph shows the cell size (FSC) and complexity (SSC) pattern of the counted cells, and the circled area of the box plot is displayed to refer to the size patterns for live cells (PI-). This graph is shown with an illustrative purpose for morphological changes, rather than the quantification of necrosis or apoptosis, which have already been shown in the previous figure. It can be comparatively observed that the population of GM-MØ changes its FSC/SSC ratios after GMC treatment. Lower or higher doses of GMC (5, 10 or 60 µg/mL) modified the results of the size and complexity analysis in a similar way. Figure 5B shows a representative contrast phase microscope picture of the macrophage left untreated or treated with 20 µg/mL GMC for 24 h, where some particles of GMC can be detected inside the cells as a consequence of phagocytic activity. 

Figure 6A,B show representative images of flow cytometry analysis and microscope pictures of GM-MØ after being untreated or treated with a lower dose of GMC (10 µg/mL) for 7 days during monocyte-to-macrophage differentiation in culture. As in Figure 5, the images show similar changes in the FSC/SSC analysis and microscope images, suggesting that macrophages phagocyte the GMC product. Lower or higher doses of GMC (5 or 60 µg/mL) modified the results of the size and complexity analysis in a similar way. 

### 3.5. Viability, Immunological Activity and Phagocytosis During GMC Treatment of M-MØ 

Regarding the effect of GMC on anti-inflammatory macrophages (M-MØ), Figure 7A,B show the necrosis and apoptosis of fully differentiated M-MØ after 24 h of GMC treatments. GMC at 20 or 60 µg/mL promoted a weak increased tendency in both necrosis and apoptosis when compared to controls, only being statistically significant for apoptosis with 60 µg/mL (Figure 7B). We also studied the viability of M-MØ when GMC was present for 7 days during the macrophage differentiation. Figure 7C,D show that, interestingly, GMC was able to reduce necrosis and apoptosis during M-MØ differentiation, being statistically significant in terms of the apoptosis reduction using 60 µg/mL GMC. No detectable levels of activin A were released into the media by M-MØ.

Figure 8A,B show representative images of flow cytometry analysis and microscope pictures of fully differentiated M-MØ after being left untreated or treated with 20 µg/mL GMC for 24 h. The images show changes in the complexity and size, suggesting that M-MØ macrophages phagocyte the GMC product. Lower or higher doses of GMC (5, 10 or 60 µg/mL) modified the results of the size and complexity analysis in a similar way. 

Figure 9A,B show representative images of flow cytometry analysis and microscope pictures of M-MØ after being untreated or treated with a lower dose of GMC (10 µg/mL) for 7 days during monocyte–macrophage differentiation in culture. As in the previous figures, the images show similar changes in the FSC/SSC analysis and microscope images, suggesting that macrophages phagocyte the GMC product. Lower or higher doses of GMC (5 or 60 µg/mL) modified the results of the size and complexity analysis in a similar way.

## 4. Discussion

To evaluate the influence of GBMs on the immune system is a critical step in their translational application at pre-clinical or clinical stages. GMC, as with CNTs and other CNFs, is a fiber-like GBM. They have beneficial advantages in biomedical applications when properly administered at low concentrations [3,4,5,6,7,8,9,10,25]. The differences between them are diameters ranging from ten to hundreds of nm and lengths from a few nm to hundreds of μm, the graphene layer arrangements (which go from a perpendicular or cone-like shape to rolled-up cylinders) and importantly, the cost of production [2,3,4,22,25]. The literature has suggested that most GBMs may interact with biomolecules, blood cells and peripheral tissues and therefore they may induce effects to some extent, varying from being inert to bioactive or even toxic. Many discrepancies have been reported about their effect on the immune response (survival, adherence, the polarization state, inflammatory activity or phagocytosis), which can be attributed to intrinsic physicochemical properties (i.e., the purity, shape, size, graphene layer arrangements) and the experimental model used (i.e., the concentration, administration pathway) [11,12,13,14,15,16,17,18,19], also taking into consideration that most of these publications are particularly focused on toxicological studies using high concentrations and invasive administrations of GBM. 

Ultrasonication is part of the procedure to ensure the range of the GMC particle size, which is around 300 nm [25]. This size is potentially capable of invading the internal body when it is presented in a suspension [32]. Because of the sonication, metal traces may be released and agglomerated in the product [5]. The reduction of metals by treating the dried GMC material with strong acid was ensured, as well as the elimination of potential endotoxin contaminants that could induce inflammation by also applying aggressive thermal treatment to the final product [11,15,16]. 

In our previous study, we observed that GMC used in suspension at a dose lower than 150 µg/mL lacked depletory effects and was able to change the phenotype of cultured adipocytes as well as in the subcutaneous adipose tissue, using a topical administration of GMC in a rodent model [25]. Moreover, other CNF-based materials are biologically functional and biocompatible, either in vitro or in vivo, when used in this concentration range [3,4,5,6,7,8,9,10,14,25]. 

Because our study is related to the direct contact of GMC with immune cells, the principal aim was to study whether GMC has toxic effects on leukocytes or triggers some inflammatory mechanisms. GMC’s potential toxicity in immune cells was determined for each progeny of white blood cells, paying special attention to the percentage of necrosis and apoptosis. We can consider several possible mechanisms of interaction according to our previous work and the literature: the GMC fiber-like structure could mimic the extracellular matrix mesh and potentially interact with extracellular receptors such as integrins or chemokine receptors as a TLR, promoting intracellular signaling which can modify the behavior of the cells. We already demonstrated that GMC could modify the behavior of systemic cells through the interaction with integrins [25], and others demonstrated that some GBMs may interact with related mediators using other cellular models to modify their differentiation [1,20,33,34,35,36]. Unfortunately, studying the kinetics of GMC is extremely difficult, because it is a carbon-based structure that is not tagged and has no intrinsic fluorescence to be traced in a biological environment [37].

GMC used in a suspension range from 10 to 60 µg/mL neither had toxic effects nor evoked inflammatory responses when applied for 24, 48 or 72 h to T- or B-lymphocyte progenies both in a steady state or under the stimulation of specific cytokines. Similarly, NK cells’ degranulation capacity in response to the target cell type was not altered after being cultured for 7 days with GMC. 

As part of the innate immune response, monocytes are recruited and polarized to phagocytic pro-inflammatory macrophages (GM-MØ) or to anti-inflammatory macrophages (M-MØ). 

Most publications on GBM-mediated immune responses have been carried out on macrophages, because they play a key role in nanoparticle uptake, removal and trafficking in vivo through phagocytosis, [5,11,12,13,14,15,16,32]. To study the inflammatory behavior of macrophages, we first used the monocyte cell line THP-1. Once THP-1 is activated with a pro-inflammatory stimulus such as TGF-β, the expression of cytokine IL-1β is increased in correlation with its release into the culture media, which did not happen when using GMC at a concentration of 20 µg/mL for 24 h [28,29].

We also studied the release of another pro-inflammatory marker, activin A, by fully differentiated GM-MØ after GMC treatment for 24 h or during 7 days of differentiation. A concentration of 10 or 20 µg/mL did not modify the release of activin A. There was a statistically significant increase in activin A released when using the highest dose of 60 µg/mL after 7 days, but not to the extent of a strong pro-inflammatory stimulus, such as the effect caused by TGF-β on THP-1 [28,29].

On the other hand, when monocytes were treated after or during their differentiation to M-MØ, no activin A was released as expected. In any case, these data demonstrate that GMC does not repolarize M-MØ to GM- MØ. 

The necrosis or apoptosis of polarized GM-MØ or M-MØ did not change after applying 10 or 20 µg/mL GMC, either after 24 h or during 7 days of differentiation. Only the highest dose, 60 µg/mL, increased the necrosis and/or apoptosis with statistical significance after 24 h in macrophages, but the percentage of both populations was still physiologically irrelevant. It is plausible to think that 60 µg/mL is the saturation dose for macrophages to phagocyte the product, as the consequence of some toxic effect after the phagocytosis detected either in the analysis of the size and complexity change in the macrophages or the observation under a microscope. However, it is important to notice that apoptosis and necrosis somehow decreased during the 7 days of treatment during M-MØ differentiation, while it was increased during GM-MØ differentiation. This could be interpreted as GMC selectively increasing the necrosis and/or apoptosis of pro-inflammatory GM-MØ in contrast with the facilitated survival of M-MØ during differentiation. 

Some studies pointed out a GBM interaction with TLRs present in the membrane of the macrophages [38,39,40]. Therefore, it is plausible that some change in the macrophage survival or immune reactivity may be driven by downstream intracellular receptors, as studied elsewhere [40,41,42,43,44,45]. Moreover, several GBMs have been studied to manipulate the polarization of GM-MØ and M-MØ [14,15,16,46,47,48] and, interestingly, in some other studies the studied GBM favored M-MØ polarization [49,50,51,52]. Nevertheless, further research may be devoted to understanding whether GMC can affect the survival or repolarization of macrophages and how this can be advantageous in terms of potential immunotherapy as suggested for other GBM products. [46,47,48,49,50,51,52]

## 5. Conclusions

The CNF-based material GMC, free of impurities and characterized to have a particle size of hundreds of nm and a diameter of tens, was biologically functional, as demonstrated in our previous publications. Here, we demonstrated that used at concentrations between 10 and 60 µg/mL for several days, it was non-toxic and did not evoke an immunological response in cultured purified human lymphocytes or NK cells, either in basal or under activating conditions. GM-MØ and M-MØ both phagocyted GMC at a concentration of 20 or less µg/mL, without evoking changes in the viability or activity. However, use at 60 µg/mL slightly increased deleterious effects in GM-MØ, although the analysis of the data reflects a weak increase that probably should be considered biologically irrelevant. Interestingly, since several GBMs have been studied to manipulate the polarization of GM-MØ and M-MØ, it is plausible that the changes in shape and even the survival of M-MØ in detriment to GM-MØ could be driven by downstream intracellular receptors that may be implicated in the macrophage’s repolarization. Altogether, these results suggest that GMC, and similar graphene-based materials, including fiber-like CNFs and CNTs, may offer potential uses in pre-clinical settings besides those already demonstrated by us, and opens up potential uses in other therapeutic strategies, since it seems to be highly biocompatible, with minimal immunological activity that will always be dependent on the dose, administration or formulation of the product. Further research may be devoted to understanding whether GMC can affect macrophages’ repolarization, pointing to GMC as an advantageous potential immunotherapy.

## Figures and Tables

**Figure 1 nanomaterials-14-01945-f001:**
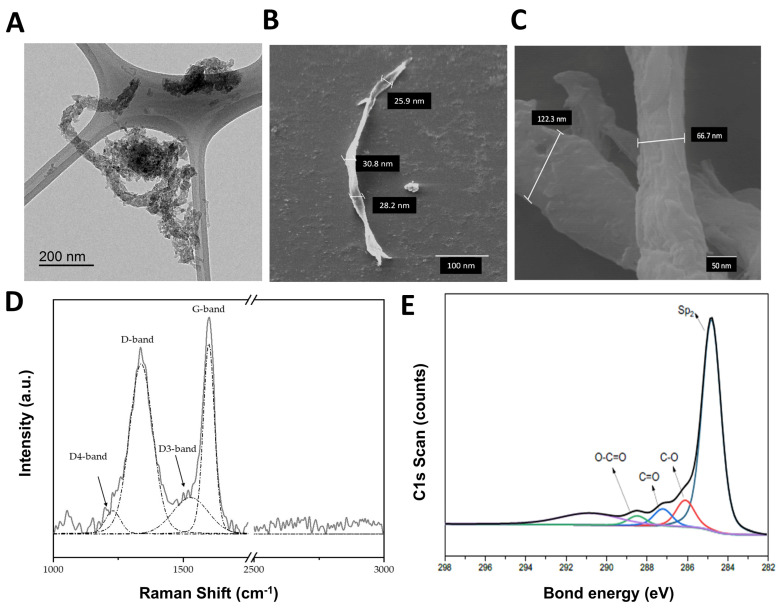
GMC characterization: Representative images of agglomerated non-suspended dry GMC obtained through (**A**) High-Resolution Transmission Electron Microscopy (TEM, scale bar 200 nm) and (**B**,**C**) Scanning Electron Microscopy (SEM, scale bars 100 and 50 nm, respectively). (**D**) Raman analysis indicating D, D3, D4 and G bands, (**E**) high-resolution spectrum of C1s obtained using X-ray photoelectron spectroscopy (XPS).

**Figure 2 nanomaterials-14-01945-f002:**
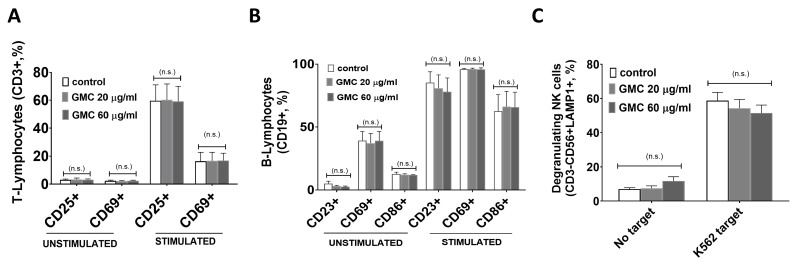
Immunological activity of different lymphocyte populations during GMC treatment. PBMCs were isolated from buffy coats of healthy donors. In some cases, when required, specific cell lineages were purified. After appropriate treatments for activation and/or GMC stimulation, cells were stained with specific surface antibodies for the active lineages and analyzed using flow cytometry. (**A**) T-lymphocytes in the context of total PBMCs were stimulated or not under culture conditions with 20 ng/mL IL2 and 0.5 μg/m PHA-M for 96 h and 20 ng/mL IL2 for another 48 h. Afterwards, cells were treated with GMC (0, 20 or 60 µg/mL) for another 24 h. Histograms represent the % of T-lymphocytes (CD3+) positive for the activation makers CD69 and CD25. (**B**) B-lymphocytes were isolated using a commercial kit and co-treated for 24 h with GMC (0, 20 or 60 µg/mL) and stimulated or not (unstimulated) with B-lymphocyte activators (1μM ODN 2395-CpG, 1 mg/mL IgG + IgM and 100 U/mL IL4). Histograms represent the % of B-lymphocytes (CD19+) positive for the activation markers CD23, CD69 and CD86. (**C**) PBMCs were cultured for 7 days with GMC (0, 20 or 60 µg/mL) and co-cultured afterwards for 2 h with NK-degranulating target cell type K562 or no target (unstimulated). Histograms represent the % of degranulating NK cells (CD3−, CD56+, LAMP1+). The results are presented as means ± SEM from *n* = 6 independent donors. No statistically significant differences were found between control and GMC treatments within any of the studied subpopulations. (n.s.) = no significant differences vs. control.

**Figure 3 nanomaterials-14-01945-f003:**
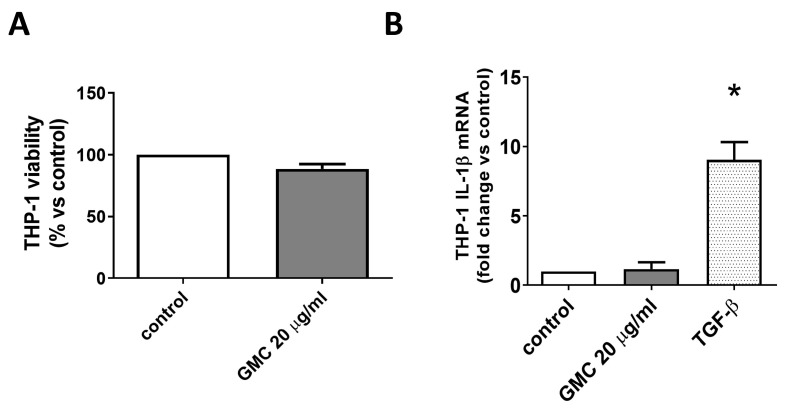
Viability and immunological activity of cultured human monocyte THP-1 cell line. THP-1 was treated with GMC (0, 20 µg/mL) for 24 h. (**A**) Viability was determined using trypan blue cellular exclusion. (**B**) IL-1β mRNA expression normalized to β-actin levels, determined using RT-qPCR. TGF-β1 (20 ng/mL) was added as an inflammatory positive control. Data are expressed as mean ± SEM from *n* = 12 experiments. * = *p* < 0.05 vs. control.

**Figure 4 nanomaterials-14-01945-f004:**
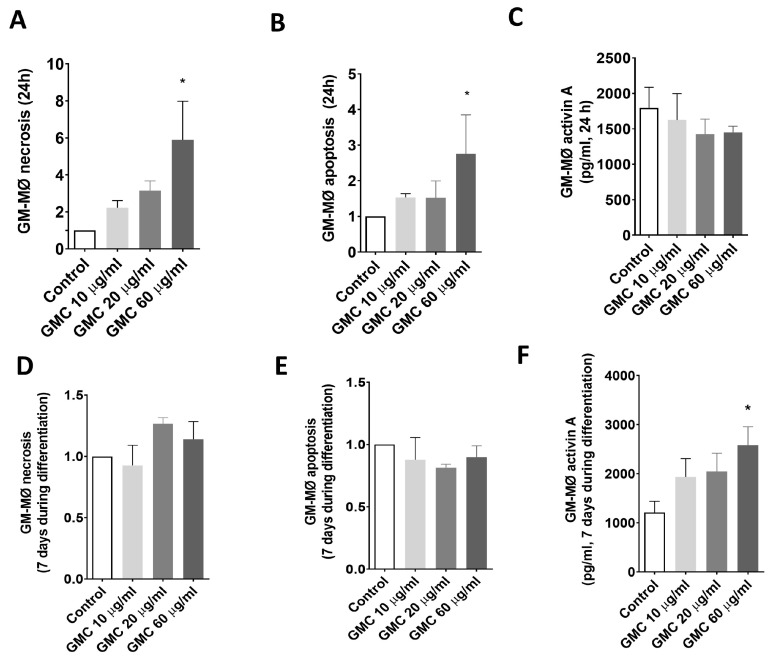
Viability and activin A production of GM-MØ during GMC treatment. (**A**) Ratios vs. the control of necrotic (Annexin V− and PI+) and (**B**) apoptotic (Annexin V+, PI−) cells and (**C**) activin A release in the supernatant (pro-inflammatory marker) from fully differentiated GM-MØ after 24 h of GMC treatments (10, 20 or 60 µg/mL) or when left untreated (control). (**D**) Ratios of necrotic, (**E**) apoptotic and (**F**) activin A levels released from GM-MØ after 7 days of GMC treatment during their macrophage differentiation. Data are expressed as mean ± SEM from *n* = 3–6 experiments. * = *p* < 0.05 vs. control.

**Figure 5 nanomaterials-14-01945-f005:**
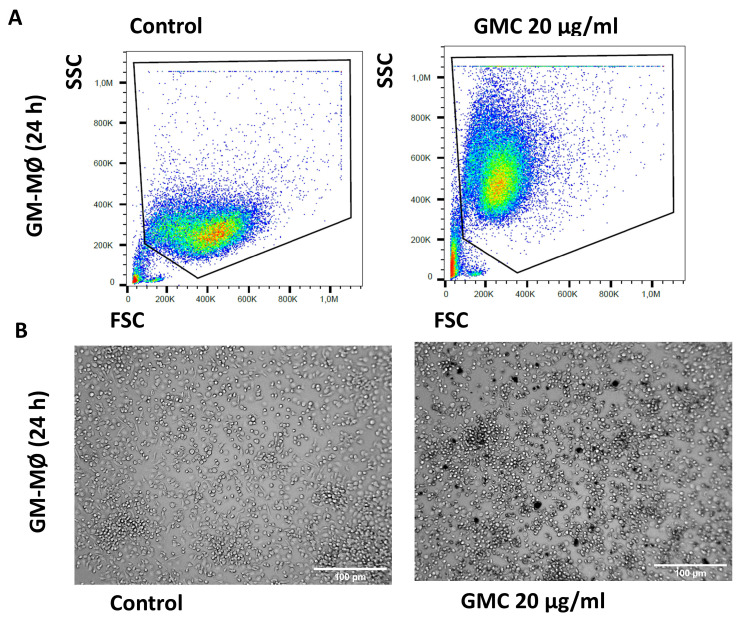
Morphological pattern and phagocytosis during GMC treatment of fully differentiated GM-MØ. (**A**) Representative flow cytometry analysis box plot for cell size (FSC) and complexity (SSC) comparison of GM-MØ after 24 h of 20 µg/mL GMC treatment or when left untreated (control). (**B**) Representative contrast phase microscope pictures of the cells after GMC treatment. Scale bars: 100 µm.

**Figure 6 nanomaterials-14-01945-f006:**
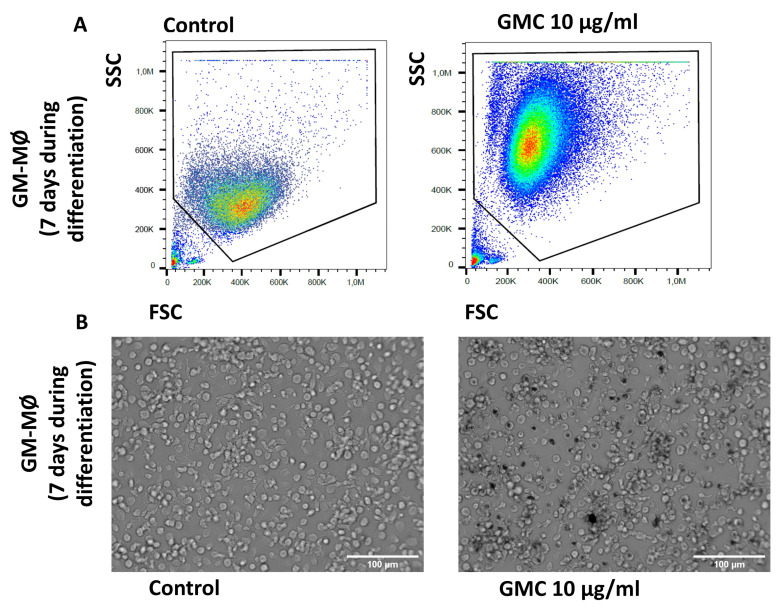
Morphological pattern and phagocytosis during GMC treatment during monocyte-to-GM-MØ differentiation. (**A**) Representative flow cytometry analysis box plot for cell size (FSC) and complexity (SSC) comparison of GM-MØ after 7 days of 10 µg/mL GMC treatment or when left untreated (control). (**B**) Representative contrast phase microscope pictures of the cells after GMC treatment. Scale bars: 100 µm.

**Figure 7 nanomaterials-14-01945-f007:**
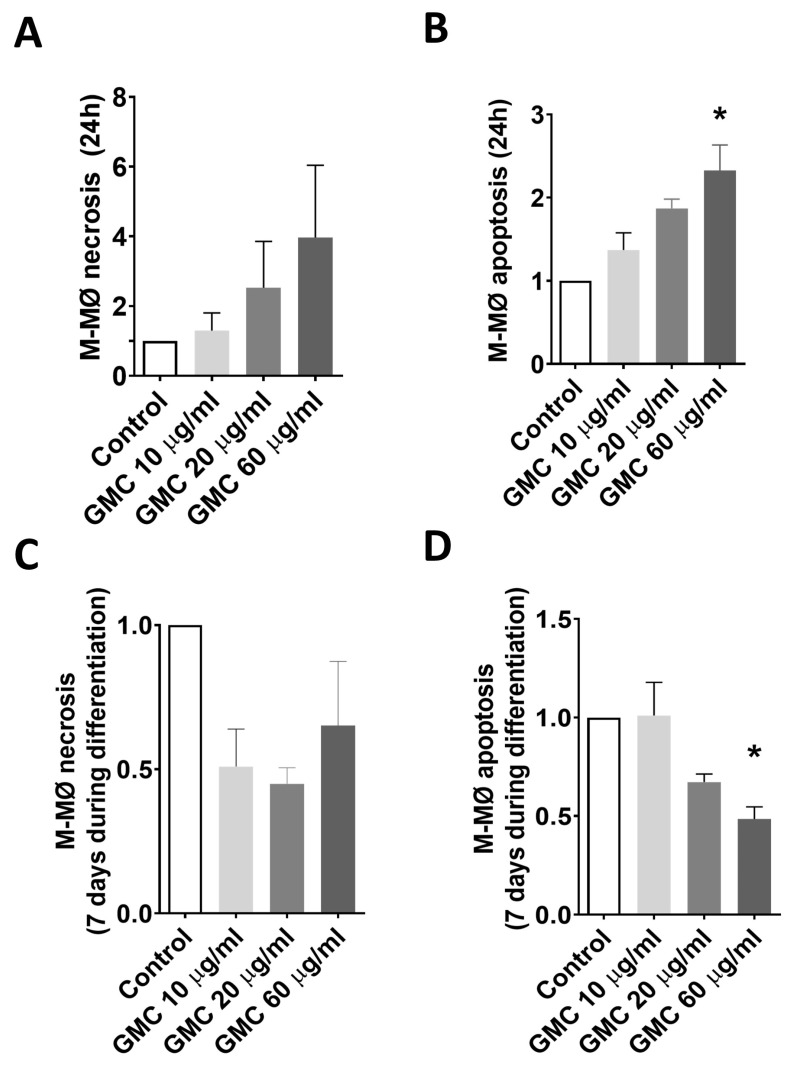
Viability of M-MØ during GMC treatment. (**A**) Ratios vs. control of necrotic (Annexin V− and PI+) and (**B**) apoptotic (Annexin V+, PI−) fully differentiated M-MØ after 24 h of GMC treatments (10, 20 or 60 µg/mL) or when left untreated (control). (**C**) Ratios of necrotic and (**D**) apoptotic M-MØ after 7 days of GMC treatment during their macrophage differentiation. Data are expressed as mean ± SEM from *n* = 3 experiments. * = *p* < 0.05 vs. control.

**Figure 8 nanomaterials-14-01945-f008:**
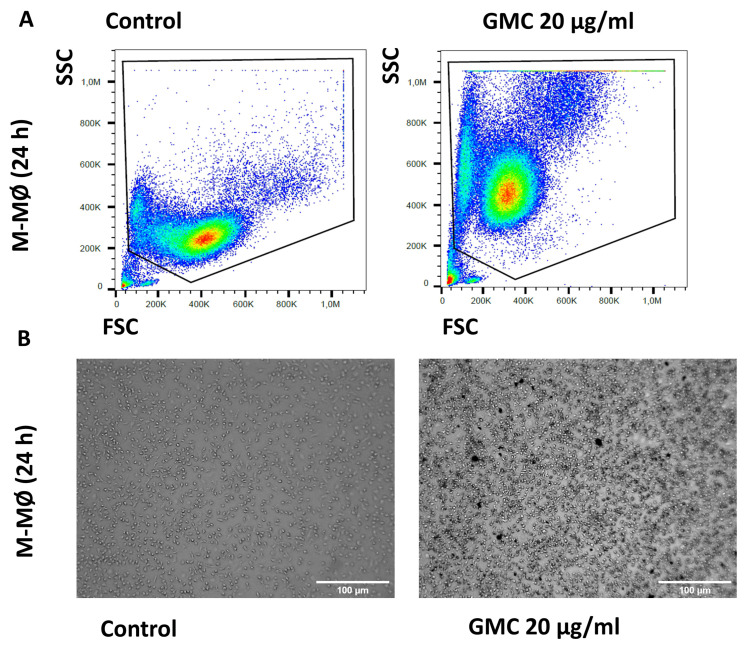
Morphological pattern and phagocytosis during GMC treatment of fully differentiated M-MØ. (**A**) Representative flow cytometry analysis box plot for cell size (FSC) and complexity (SSC) comparison of M-MØ after 24 h of 20 µg/mL GMC treatment or when left untreated (control). (**B**) Representative contrast phase microscope pictures of the cells after GMC treatment. Scale bars: 100 µm.

**Figure 9 nanomaterials-14-01945-f009:**
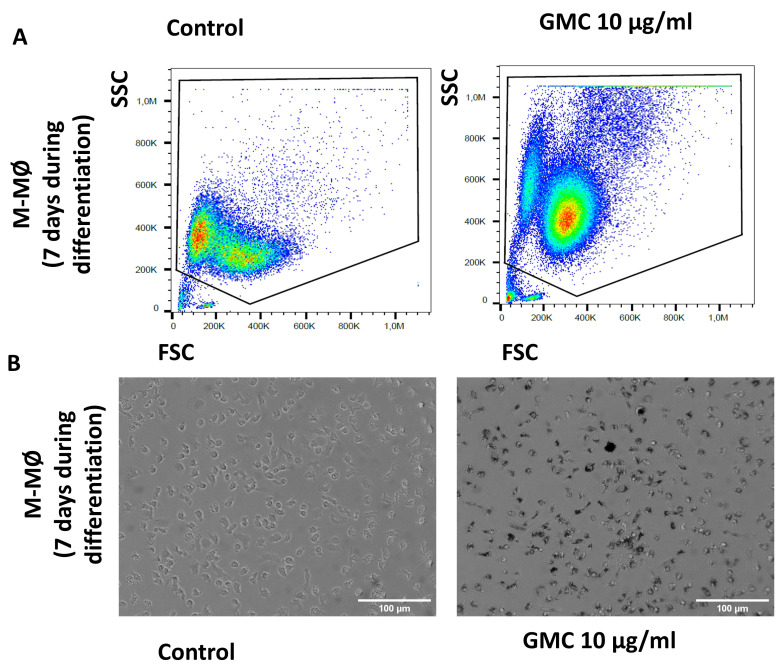
Morphological pattern and phagocytosis during GMC treatment during M-MØ differentiation. (**A**) Representative flow cytometry analysis box plot for cell size (FSC) and complexity (SSC) comparison of M-MØ after 7 days of 10 µg/mL GMC treatments or when left untreated (control). (**B**) Representative contrast phase microscope pictures of the cells after GMC treatment. Scale bars: 100 µm.

**Table 1 nanomaterials-14-01945-t001:** PBMCs’ viability after treatment with GMC.

% of Total PBMCs	Control	20 µg/mL	60 µg/mL
G0/G1 phase	90.3 ± 2.0	92.5 ± 1.3	93.5 ± 0.7
G2/M phase	3.4 ± 0.2	3.3 ± 0.8	2.3 ± 0.5
S phase	0.9 ± 0.1	1.1 ± 0.2	1.2 ± 0.3
Apoptosis	3.1 ± 1.0	1.9 ± 0.4	2.1 ± 0.5
Necrosis	4.9 ± 1.8	2.7 ± 0.6	2.8 ± 0.4

Human PBMCs were isolated from the buffy coats of healthy donors and left untreated (control) or treated with GMC (20 or 60 µg/mL) for 24 h, then stained afterwards with propidium iodide (PI) and analyzed using flow cytometry. Necrotic, apoptotic and viable cells at different cellular phases are represented as % of total PBMC count. Values are means ± SEM from *n* = 6 independent donors.

## Data Availability

Data are contained within the article. The original contributions presented in the study are included in the article; further inquiries can be directed to the corresponding author.

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
