# Peer review of "A Graphene-Based Bioactive Product with a Non-Immunological Impact on Mononuclear Cell Populations from Healthy Volunteers"

_nanomaterials, 2024, doi:10.3390/nano14231945_

Round 1

Reviewer 1 Report

Comments and Suggestions for Authors

The aim of the manuscript entitled "A graphene-based bioactive product with non-immunological impact in mononuclear cell populations from healthy volunteers" is to evaluate the immunoreactivity and safety of GMC, a graphene-based nanomaterial derived from carbon nanofibers, when introduced into cultured human lymphocytes, macrophages (both inflammatory and reparative types), and monocytes. By assessing cellular viability, necrosis, apoptosis, and the production of key inflammatory markers, the study aims to determine the biological compatibility of GMC at different concentrations (20 and 60 µg/ml) under various basal and activated conditions, with a focus on its potential utility in living systems.

The manuscript is within the journal's scope and is generally well-prepared. The topic is highly relevant. The findings will contribute to understanding GMC's immunological safety and potential biomedical applications. The manuscript is well prepared in general. Still, some issues need to be addressed. My specific comments are given below.

The abstract implies the goal is to assess the immunoreactivity and safety of GMC, but the aim is not clearly stated. Moreover, the conclusion of the abstract is superficial, stating that GMC is "immunologically safe" without explaining the broader implications of this finding or acknowledging any potential limitations.

The introduction should be more informative, as the context is not obvious, and the motivation for this study needs to be more clearly emphasized. It is crucial to explain why investigating the impact of GMC on immune cells is important. Providing more background on the significance of GMC's potential biomedical applications and the specific relevance of its immunoreactivity could help establish a clearer rationale for the study. Additionally, highlighting the current gaps in knowledge and the potential implications for therapeutic use or safety assessment would be beneficial.

A high level of plagiarism has been detected in the description of methods in Section 2. This needs to be corrected by rephrasing the text and ensuring original wording is used while appropriately citing any referenced sources.

Abbreviations for characterization methods such as DLS, TEM, and SEM should be introduced where the method is first mentioned (Section 2) and then used consistently throughout the manuscript.

In Figure 1D, the Raman spectra have not been deconvoluted. It should be decomposed into 5 peaks before comparing the intensities of the D (ID) and G (IG) bands. This will provide a more accurate and detailed analysis of the spectral components.

The first sentence of the Discussion should be changed to "To evaluate the influence of GBM on the immune system is a critical step in its translational application at pre-clinical or clinical stages."

The conclusion is very brief and does not provide a deeper critical reflection; it simply repeats what has already been stated in the manuscript. This section should be improved by offering a more comprehensive analysis of the findings discussing their broader implications, potential limitations, and suggestions for future research.

Author Response

The aim of the manuscript entitled "A graphene-based bioactive product with non-immunological impact in mononuclear cell populations from healthy volunteers" is to evaluate the immunoreactivity and safety of GMC, a graphene-based nanomaterial derived from carbon nanofibers, when introduced into cultured human lymphocytes, macrophages (both inflammatory and reparative types), and monocytes. By assessing cellular viability, necrosis, apoptosis, and the production of key inflammatory markers, the study aims to determine the biological compatibility of GMC at different concentrations (20 and 60 µg/ml) under various basal and activated conditions, with a focus on its potential utility in living systems.

The manuscript is within the journal's scope and is generally well-prepared. The topic is highly relevant. The findings will contribute to understanding GMC's immunological safety and potential biomedical applications. The manuscript is well prepared in general. Still, some issues need to be addressed. My specific comments are given below.

1.- The abstract implies the goal is to assess the immunoreactivity and safety of GMC, but the aim is not clearly stated. Moreover, the conclusion of the abstract is superficial, stating that GMC is "immunologically safe" without explaining the broader implications of this finding or acknowledging any potential limitations.

RESPONSE 1: We thank the reviewer for highlighting the need for broader knowledge in the abstract. We have revised it to include additional information on the potential applications of GMC and the implications of our research findings. All the changes are in red font.

2.- The introduction should be more informative, as the context is not obvious, and the motivation for this study needs to be more clearly emphasized. It is crucial to explain why investigating the impact of GMC on immune cells is important. Providing more background on the significance of GMC's potential biomedical applications and the specific relevance of its immunoreactivity could help establish a clearer rationale for the study. Additionally, highlighting the current gaps in knowledge and the potential implications for therapeutic use or safety assessment would be beneficial.

RESPONSE 2: We appreciate the feedback on improving the introduction. Additionally, we thoroughly reviewed the manuscript to ensure grammatical accuracy.

3.- A high level of plagiarism has been detected in the description of methods in Section 2. This needs to be corrected by rephrasing the text and ensuring original wording is used while appropriately citing any referenced sources.

RESPONSE 3: To improve overall clarity and maintain originality, we rephrased the text where possible. In the methods section, however, originality is challenging due to the reliance on established protocols commonly used in the field. Nevertheless, we made efforts to revise the language, incorporating citations (including self-citations) where appropriate.

4.- Abbreviations for characterization methods such as DLS, TEM, and SEM should be introduced where the method is first mentioned (Section 2) and then used consistently throughout the manuscript.

RESPONSE 4: We added the corresponding abbreviations in the methods section, following your suggestion.

5.- In Figure 1D, the Raman spectra have not been deconvoluted. It should be decomposed into 5 peaks before comparing the intensities of the D (ID) and G (IG) bands. This will provide a more accurate and detailed analysis of the spectral components.

RESPONSE 5: Thank you for the observation regarding the deconvolution of the Raman spectra .Our analysis approach focuses on comparing the intensities of the D (ID) and G (IG) bands without deconvolution to retain the overall spectral profile, obtaining the main information on the degree of defects of the material that GMC presents which could affect its final properties. While deconvolution could potentially offer more detailed spectral information, our goal here was to capture broader trends and relationships, which we believe are well-represented without additional spectral decomposition.

This method aligns with previous studies in our field, from us and others, using the same raw material o similar CNF-based materials as the product GMC, where direct intensity measurements of the D and G bands have been shown to provide valuable insights. We did mention some of these works in our previous manuscript (Salesa B, et al.  Biomedicines. 2021, PMID: 34572341; de Frutos S, Biomater Sci. 2023 , PMID: 37306667) and now we added another reference from our group ( Romero, A. et al. Effects of oxidizing procedures on carbon nanofibers surface and dispersability in an epoxy resin. Mater. Chem. Phys. 2020, 243, 122571.)

6.- The first sentence of the Discussion should be changed to "To evaluate the influence of GBM on the immune system is a critical step in its translational application at pre-clinical or clinical stages."

RESPONSE 6: Following your suggestion, we modified the text.

7.- The conclusion is very brief and does not provide a deeper critical reflection; it simply repeats what has already been stated in the manuscript. This section should be improved by offering a more comprehensive analysis of the findings discussing their broader implications, potential limitations, and suggestions for future research.

RESPONSE 7: We also addressed the reviewer's suggestion for a deeper analysis in the conclusion. The revised section emphasizes the significance of our findings regarding GMC and its potential applications in preclinical studies.

Reviewer 2 Report

Comments and Suggestions for Authors

The article proposed describe the biological analysis of the application of a carbon nanomaterial, GMC, on specific kind of cells from healthy volunteers. The authors extend the study on the biological compatibility of a carbon based material produced by the firm to which some of the co-authors belong. From the chemical point of view there is no novelty but the work presents many different analyses showing the interesting and useful properties of the material. So I believe that the work deserve to be published in the present form. One doubt: the name GMC is a name that comes from the firm? Is it a registered trademark? In this case it should be highlighted. Otherwise the name should be explained.

Author Response

The article proposed describe the biological analysis of the application of a carbon nanomaterial, GMC, on specific kind of cells from healthy volunteers. The authors extend the study on the biological compatibility of a carbon based material produced by the firm to which some of the co-authors belong. From the chemical point of view there is no novelty but the work presents many different analyses showing the interesting and useful properties of the material. So I believe that the work deserve to be published in the present form. One doubt: the name GMC is a name that comes from the firm? Is it a registered trademark? In this case it should be highlighted. Otherwise the name should be explained.

RESPONSE: We appreciate the reviewer’s positive comments. GMC refers to a CNF-based material manufactured and patented by Graphenano Medical Care S.L. (WO 2020/016319). While it has no registered trademark, we retained the nomenclature used in our previous publication (de Frutos et al., Biomater. Sci., 2023, 11(14), 4916–4929), cited in the manuscript. Additional details on these findings and potential applications have been included.

Reviewer 3 Report

Comments and Suggestions for Authors

Journal: nanomaterials-

MS Id: 3291980-peer-review-v1

Title: A graphene-based bioactive product with non-immunological impact in mononuclear cell populations from healthy volunteers

This study aimed to assess the immunoreactivity of the compound, considering its potential applications in living systems, by introducing it to cultured lymphocytes and macrophages derived from pooled peripheral blood mononuclear cells (PBMCs) from healthy donor. The article is clearly structured and effectively written. However, some issues still need to be addressed and revised before the paper can be accepted.

Specific comments:

1.     Please include the IRB approval number in Section 2.2, 'Leukocyte Progeny Purification Culture Conditions and Analysis.

2.     Please specify the country and city of origin for all experimental reagents in the 'Materials and Methods' section.

3.     Please include additional details in Section “2.3, 'Apoptosis, Necrosis and Cell Cycle Profiles.”

4.     In the Results section, please change the first instance of "During" to lowercase, "during”.

5.     Please revise the caption for Figure 1 to ensure consistency with the description provided in the “”Results section”.

6.     Please delete this section, as it is not essential to the results. “GMC physicochemical characteristics have been previously published by the group [26]. Briefly, GMC Z-average size was limited inside a range close to 300 nm as determined by Dynamic Light Scattering (DLS) technique, with a main first peak of hundreds of nm (661.9 nm + st. dev 387.5, approximately 90 % of total particles) and a secondary peak of thousands (5093 nm + st. dev 827.5, approximately 10 % of the particles).”

7.     Please include IL-1b ELISA data to confirm that IL-1b mRNA is translated into protein and actively secreted by immune cells, as depicted in Figure 3.

8.     Please explain why there was no significant effect observed after treatment with the GMC functional concentration (20 μg/ml) in Figures 4A and 4B.

9.     In the Results section, please explain why Figure 4F shows that GMC at 20 μg/ml did not induce an increase in Activin A expression, while a concentration of 60 μg/ml resulted in a slight increase?

10.  Please provide the flow cytometry quantification data for GM-MØ in Figure 5A after 24 hours of GMC treatment at concentrations of 0 and 20 μg/ml, as well as for Figures 6, 8, and 9.

11.  Please explain why there was no significant difference observed after 24 hours of GMC treatment at 0, 20, or 60 μg/ml in Figure 7A, as there appears to be an effect.

12.  Please include scale bars in Figures 5B, 6B, 8B, and 9B.

Comments on the Quality of English Language

none

Author Response

This study aimed to assess the immunoreactivity of the compound, considering its potential applications in living systems, by introducing it to cultured lymphocytes and macrophages derived from pooled peripheral blood mononuclear cells (PBMCs) from healthy donor. The article is clearly structured and effectively written. However, some issues still need to be addressed and revised before the paper can be accepted.

Specific comments:

  1. Please include the IRB approval number in Section 2.2, 'Leukocyte Progeny Purification Culture Conditions and Analysis.

RESPONSE 1: Thank you for your observations. We also realized that the paragraph refering to the ethical statements in the section 2.2 was confusing and not completely correct. We change the text to include a more detailed information as follow:

“… Anonymized Buffy coats from a total of 6 anonymous healthy donors were gently donated by the Blood bank at the Madrid Transfusion Centre of Autonomous Madrid Government to the National Centre for Biotechnology (CSIC, IRB approval number 280508). The procedures to obtain blood and buffy coats followed the standards of transfusion institutions, including the informed consent from the donors. Human peripheral blood mononuclear cells (PBMC) were isolated from donated buffy coats sam-ples using Lymphoprep separation (Nycomed Pharma AS, Oslo, Norway).”

  1. Please specify the country and city of origin for all experimental reagents in the 'Materials and Methods' section.

RESPONSE 2: We completed the origins from our reagents in the Materials Section

  1. Please include additional details in Section “2.3, 'Apoptosis, Necrosis and Cell Cycle Profiles.”

New details have been incorporated in material and methods section, as well as a new paragraph describing the mechanisms that the use of annexin V and PI facilitates the determination of necrosis and apoptosis. Annexin V binds to phosphatidylserine, a phospholipid that translocates from the inner to the outer leaflet of the cell membrane during early apoptosis. PI, on the other hand, intercalates with DNA but can only enter cells with compromised plasma membranes. Cells were analysed by flow cytometry to differentiate between viable, early or late apoptotic, and necrotic cells depending on the dyed, detected in the flow cytometry.

  1. In the Results section, please change the first instance of "During" to lowercase, "during”.

RESPONSE 4: The text have been changed.

  1. Please revise the caption for Figure 1 to ensure consistency with the description provided in the “”Results section”.

RESPONSE 5: Thank you to the reviewers for noticing that the relation between the figure caption and the description was confusing. We changed it.

  1. Please delete this section, as it is not essential to the results. “GMC physicochemical characteristics have been previously published by the group [26]. Briefly, GMC Z-average size was limited inside a range close to 300 nm as determined by Dynamic Light Scattering (DLS) technique, with a main first peak of hundreds of nm (661.9 nm + st. dev 387.5, approximately 90 % of total particles) and a secondary peak of thousands (5093 nm + st. dev 827.5, approximately 10 % of the particles).”

RESPONSE 6: Since two of the reviewers considered that the DLS information is not essential for the results section, we eliminated it. DLS characterization was performed also in our previous work. Thus, we do mention the cite withing the characterization results section.

  1. Please include IL-1b ELISA data to confirm that IL-1b mRNA is translated into protein and actively secreted by immune cells, as depicted in Figure 3.

According to an extent literature, the mRNA expression of the pro-inflammatory cytokine IL-1β by THP1 macrophages, under stimulatory conditions in vitro correlates with its release to the culture media. Between the stimuli that may increase expression and release of IL-1β, we used TGF-β as a positive control.

Some recent publications that related IL-1β RT-qPCR with ELISA in THP-1 after inflammatory stimuli are :

1.- Jiang W et al, Theranostics. 2024 Jan 1;14(3):1049-1064. PMID: 38250043;.

2.- Medeiros R et al, Molecules. 2021 Jun 18;26(12):3721. PMID: 34207168;.

3.- Gopinath VK et al.m Mohammad MG, Int Endod J. 2023 Jan;56(1):27-38. PMID: 36190353.

Moreover, the following cites show the expression in THP-1 of IL-1β by RT-qPCR (but not its release), even using TGF-β as the inflammatory stimulus, similar to our settings. For that reason, we decided to add them as part of our bibliograpy.

1.- Souza ILM, et al. Int J Mol Sci. 2024 Jul 11;25(14):7605. PMID: 39062847;

2.- Poirier SJ et al. Prostaglandins Leukot Essent Fatty Acids. 2020 PMID: 32120263. 

Regarding the ELISA for IL-1β, we chose not to include data for GMC-treated THP-1 cells as we demonstrated the absence of expression at 20 µg/mL GMC. As a positive control, TGF-β significantly increased IL-1β expression, validating our assay. Instead, we focused on subsequent figures, which show that GMC-treated human macrophages released activin-A, as detected by ELISA, further supporting our conclusion that 20 µg/mL GMC does not enhance macrophage inflammatory activity, consistent with THP-1 findings.

  1. Please explain why there was no significant effect observed after treatment with the GMC functional concentration (20 μg/ml) in Figures 4A and 4B.

RESPONSE 8: We thanks the reviewer for noticing that figures 4A and 4B about necrosis and apoptosis need a more detailed explanation.

We have now re-analized the statistics of the normalized data from 0, 10, 20 and 60 μg/ml of GMC for 24h in GM-MØ and M-MØ and for 7 days during macrophage differentitations to GM-MØ and M-MØ. We now used Friedman analysis followed by Dunn’s post hoc test for multiple comparisons, since our data involves more that two groups that are paired and non-parametric. We add the new figures and statistics. Figure 4A and 4B show necrosis and apoptosis in GM-MØ after 24 h of GMC treatments. When compared to control, GMC promoted a weak increased tendency in a dose dependent manner in both necrosis and apoptosis, being statistically significant only when using the highest dose, 60 µg/ml, but differences are still not significant when using 20 µg/ml.

The p-values between control and 20 µg/ml GMC was 0.20 in 4A and >0.99 in 4B. In this case, despite the apparent disparities in the data, the statistical test fails to yield sufficient evidence to reject the null hypothesis, and therefore we need to apply it, concluding that there are no significant differences between these 2 groups.

Moreover, when taking a look to the raw data, they change from 0.61% in to 2.34% in necrosis and from 12.93% to 14.30% in apoptosis between controls and 20 µg/ml GMC-treated cells, respectively, being an increase that probably should be considered biologically irrelevant.

  1. In the Results section, please explain why Figure 4F shows that GMC at 20 μg/ml did not induce an increase in Activin A expression, while a concentration of 60 μg/ml resulted in a slight increase?

RESPONSE 9: We thanks the reviewer for noticing that figure 4F about Activin A shown need a more detailed explanation.

We now re-analized the statistics of the normalized datas from 0, 10, 20 and 60 μg/ml of GMC for 24h in GM-MØ and M-MØ and for 7 days during macrophage differentitations to GM-MØ and M-MØ. As it was explained in the last question for the referee, we now used Friedman analysis followed by Dunn’s post hoc test for multiple comparisons, since our data involves more that two groups that are paired and non-parametric. Figure 4F shows that Activin A levels were slightly increased in a GMC dose dependent manner, being statistically significant when using the highest dose, 60 µg/ml, but differences are still not significant when using 20 µg/ml.

The p-value is 0.55 between control vs 20 µg/ml GMC. Again, despite the apparent disparities in the data, the statistical test fails to yield sufficient evidence to reject the null hypothesis, and therefore we need to apply it, concluding that there are no significant differences between these 2 groups.

  1. Please provide the flow cytometry quantification data for GM-MØ in Figure 5A after 24 hours of GMC treatment at concentrations of 0 and 20 μg/ml, as well as for Figures 6, 8, and 9.

RESPONSE 10: The mentioned panels are illustrative rather than quantitative. They are intended to highlighting the alterations in macrophage morphology and size after GMC treatment, as a result of GMC phagocytosis by macrophages. We do not intend to extract any viability data from these representative dot plots, because viability data (apoptosis and necrosis) are already presented in the previous figures 4 and 7. To avoid confusion and facilitate the reading fo the figure, we also removed any number or caption that appeared inside each image.

  1. Please explain why there was no significant difference observed after 24 hours of GMC treatment at 0, 20, or 60 μg/ml in Figure 7A, as there appears to be an effect.

RESPONSE 11: In accordance with this commentary, we have now re-analized the statistics of the normalized data from 0, 10, 20 and 60 μg/ml of GMC for 24h in GM-MØ and M-MØ and for 7 days during macrophage differentitations to GM-MØ and M-MØ. As it was explained in the previous questions for the referee, we now used Friedman analysis followed by Dunn’s post hoc test for multiple comparisons, since our data involves more that two groups that are paired and non-parametric and include these data in the revised figures. Figure 7A shows necrosis of M-MØ after 24 h of GMC treatments. 20 or 60 µg/ml GMC promoted a weak increased tendency in both necrosis when compared to controls. The p-value were >0.99 between control and 20 µg/ml GMC or 60 µg/ml GMC, thus no significant. Despite the apparent disparities in the data, the statistical test failed to yield sufficient evidence to reject the null hypothesis, and therefore we applied it, concluding that there were no significant differences between these 2 groups. Moreover, when  evaluate the raw data, the median change from 2.07% (control) to 3.35% (20 µg/ml GMC) and to 5.69% (60 µg/ml GMC) of necrosis, being an increase that probably is biologically irrelevant.

In conclusion, we can assure that at low levels (10 and 20 µg/ml) of GMC, the data show no statistical significances in the increses or decreases of apoptosis, necrosis or activin A release (the latter in the case of pro-inflammatory GM-MØ). We tried to explain that the highest dose of 60 µg/ml GMC probably is producing some deleterious effect, statistically significant but weak, because the phagocitosis of the product (since figures 5, 6, 8, and 9 show the changes in morphology and the phagocyted product under each condition).

However, we also highlighted the interesting tendency of reducing apoptosis and necrosis in M-MØ during their differentiation for 7 days. This could be interpreted as GMC selectively increase necrosis and/or apoptosis of differentiating pro-inflammatory GM-MØ in contrast with a facilitated survival when differentiating to M-MØ. This explanation may be in accordance of other GBM studied that favored the M-MØ polarization (citations were originally included in the previous draft). We tried to include this explanation as part of the discussion section.

  1. Please include scale bars in Figures 5B, 6B, 8B, and 9B.

RESPONSE 12: Scale bars have been included in all the microscope images.

Reviewer 4 Report

Comments and Suggestions for Authors

The manuscript investigates the immunoreactivity of a graphene-based nanomaterial derived from carbon nanofibers for its potential application in biological systems, specifically by introducing it to cultured lymphocytes and macrophages. Cell viability (including necrosis and apoptosis) and immunoreactivity were assessed using flow cytometry, along with additional analytical techniques such as ELISA and RT-qPCR. The results show that bifunctional concentrations of 20 or 60 µg/ml of GMC do not impact lymphocyte viability over a period of one day or longer. Moreover, GMC is found to be immunologically safe as a feedstock, even at low functional concentrations. While the manuscript provides valuable information, it has several issues that need to be addressed.

1. The use of first-person pronouns should be avoided in research papers to maintain a formal and objective tone.

2. The introduction requires clarification and logical strengthening, and the manuscript contains several grammatical errors that need correction.

3. For the XPS analysis, the method used to derive carbon and oxygen content from the C 1s high-resolution spectra should be clarified.

4. The DLS results lack proper citations and do not include relevant data.

5. The scale in Figure 1A is unclear and should be specified.

6. The data presented in Figure 2 have not been analyzed for statistical significance.

7. None of the microscope images in the manuscript include magnification labels, which should be added for clarity.

8. The authors are requested to carefully review and correct any formatting errors in the references.

Author Response

The manuscript investigates the immunoreactivity of a graphene-based nanomaterial derived from carbon nanofibers for its potential application in biological systems, specifically by introducing it to cultured lymphocytes and macrophages. Cell viability (including necrosis and apoptosis) and immunoreactivity were assessed using flow cytometry, along with additional analytical techniques such as ELISA and RT-qPCR. The results show that bifunctional concentrations of 20 or 60 µg/ml of GMC do not impact lymphocyte viability over a period of one day or longer. Moreover, GMC is found to be immunologically safe as a feedstock, even at low functional concentrations. While the manuscript provides valuable information, it has several issues that need to be addressed.

  1. The use of first-person pronouns should be avoided in research papers to maintain a formal and objective tone.

RESPONSE 1: Thanks for your observations. Unless the sentence was referring to our previous works and results, we tried to modify the rest of the text to maintain an objective tone

  1. The introduction requires clarification and logical strengthening, and the manuscript contains several grammatical errors that need correction.

RESPONSE 2: We appreciate the feedback on improving the introduction. Additionally, we thoroughly reviewed the manuscript to ensure grammatical accuracy.

  1. For the XPS analysis, the method used to derive carbon and oxygen content from the C 1s high-resolution spectra should be clarified.

RESPONSE 3: Since the process of obtaining carbon and oxygen values may not be clear, an attempt will be made to explain it. The method performed for obtaining the carbon and oxygen content for the XPS spectra goes through different mathematical processes that will be explained as follows: In a first step we characterized the XPS peaks using the Gaussian-Lorentizan model, knowing that those peaks could contain overlapping peaks. The Gauss- Lorentzian peak fitting is employed to separate the overlapping peaks, allowing for the identification of specific bonding states.

After obtaining the deconvolution of the peaks, they are compared with the theoretical binding energy values of each of the functional groups that can be found in the C1s peak. The most common functional groups are C-C (Aliphatic): ~284.8 eV ; C-O: ~286.5 eV; C=O: ~288 eV ; O-C=O: ~289-290 eV

Once the functional groups in each deconvoluted peak have been identified, a quantitative study of the peak areas was performed knowing that each deconvoluted peak represents the relative abundance of each carbon chemical state. By integrating the areas of these peaks, we can quantitatively obtain the carbon structure of the material and the relation between

Finally, the quantitative data from the peak areas are normalized against the sensitivity factors for carbon and oxygen to determine the atomic concentration. This approach provides an accurate measure of the relative carbon and oxygen content in the material based on the C 1s spectra.

In addition, the article has been modified to include this information:

“The atomic % of carbons and oxygens are 79.36 and 2.51, respectively. The figure shows that GMC is composed mostly by sp2 type carbon bonds (69.63% bonds) and a small contribution of other C-to-O bonds (% C–O 9.22; C=O 5.68; O–C=O 3.15). These values have been obtained by using the Gaussian-Lorentzian peak fitting combined with a normalized peak area model to identify and quantify the carbon and oxygen groups.”

  1. The DLS results lack proper citations and do not include relevant data.

RESPONSE 4: Since two of the reviewers considered that the DLS information is not essential for the results section, we eliminated it. DLS characterization was performed in our previous work (de Frutos, S et al. Biomater. Sci. 2023, 11(14), 4916-4929.), htus we keep the citation within the characterization results section.

  1. The scale in Figure 1A is unclear and should be specified.

RESPONSE 5: Thank you for noticing that the scale in Figure 1A is not clearly visible. We replaced the image with a fully visible scale.

  1. The data presented in Figure 2 have not been analyzed for statistical significance.

RESPONSE 6: We have now clarified that statistical comparisons were performed between GMC-treated and control groups within each cell type (unstimulated and stimulated), showing no significant differences. The figure and legend have been updated to indicate “n.s.” (not significant), and this clarification has been included in the results section. We did not compare unstimulated versus stimulated conditions, as the stimulation was expected to increase the number of activated cells by design. The key finding remains that no significant differences were observed between GMC-treated and control groups within each group of cells.

  1. None of the microscope images in the manuscript include magnification labels, which should be added for clarity.

RESPONSE 7: The pictures with the corresponding scale bars have been added.

  1. The authors are requested to carefully review and correct any formatting errors in the references.

RESPONSE 8: Thank you for noticing this edition problem. We now change the format according to the specified instructions for authors in the Editorial webpage.

Round 2

Reviewer 1 Report

Comments and Suggestions for Authors

The authors addressed some of my comments properly, but not all. It is unacceptable that the Raman spectra were not deconvoluted. There are freely available resources online, including tutorials and software, that can be used to perform this analysis. Additionally, the conclusion still requires improvement. Section 2 is still not original according to the plagiarism check report. 

Author Response

Comments and Suggestions for Authors

The authors addressed some of my comments properly, but not all. It is unacceptable that the Raman spectra was not deconvoluted. There are freely available resources online, including tutorials and software, that can be used to perform this analysis. Additionally, the conclusion still requires improvement. Section 2 is still not original according to the plagiarism check report. 

RESPONSES:

We thank the reviewer for its comments:

1.- We followed the suggestion of the reviewer and added a deconvoluted Raman spectra, now in NEW FIGURE 1D. We explained the peaks observed in the results section.

2.- As suggested by the reviewer, we tried our best to rearranged some comments and perspectives in the conclusion section, which did not appear in the previous conclusion section to avoid some redundancy, since they were commented in the discussion.

3.- We tried in the first round of reviews to revise and rephrase the text in section 2, METHODS, to assure original wording, using synonyms and incorporating missing citations (including self-citations) where appropriate, as suggested by this reviewer.  

In this second review the Referee is still asking for new rephrasing to avoid “plagiarism”.

Section 2 is “methods”: a method is a normalized protocol to perform experiments. It is something stated, defined to be reproducible for the rest of the scientific community.

Plagiarism is the “practice of using another person's ideas or work and pretending that it is your own”, which can be applied in an original writing or idea but cannot be applied in technical protocols.

From our experience (we have almost 200 papers published in peer-reviewed scientific journals combining the citations from all authors of the present manuscript), method sections always “repeat” themselves, in terms of repeating inevitable sentences and words.

Most of the methods followed here are normalized procedures used as canonical worldwide (e.g., the procedure of a RT-qPCR), and it will always appear similarly written here or in related publications that use the same protocol. Sometimes, some protocols have some specificities that make them not so worldwide known, then we (and the rest of scientific community) cite the source of the original protocol. And in both cases, we added citing when appropriate through that section.

Therefore, we cannot twist more the wording to explain “differently” a normalized protocol. We ask both editor and the specific referee to consider and understand our explanation to resolve this issue.

Reviewer 3 Report

Comments and Suggestions for Authors

The authors adequately addressed all comments.were adequately responed all comments. 

Comments on the Quality of English Language

none

Author Response

Comments and Suggestions for Authors

The authors adequately addressed all comments.were adequately responed all comments. 

Thank you very much for your consideration during the reviewing process

Reviewer 4 Report

Comments and Suggestions for Authors

accept

Author Response

Comments and Suggestions for Authors

accept

RESPONSE:

Thank you very much for your consideration during the reviewing process

Round 3

Reviewer 1 Report

Comments and Suggestions for Authors

The authors have satisfactorily addressed the requested revisions. If the similarity level in Section 2 does not pose an issue for the journal or the Editor, it is acceptable from my perspective.